# Disentangling age, gender, and racial/ethnic disparities in multiple myeloma burden: a modeling study

John H. Huber [1] ✉, Mengmeng Ji [1], Yi-Hsuan Shih [2], Mei Wang[1], Graham Colditz [1] & Su-Hsin Chang [1]

Multiple myeloma (MM) is a hematological malignancy that is consistently preceded by an asymptomatic condition, monoclonal gammopathy of undetermined significance (MGUS). Disparities by age, gender, and race/ethnicity in both MGUS and MM are well-established. However, it remains unclear whether these disparities can be explained by increased incidence of MGUS and/or accelerated progression from MGUS to MM. Here, we fit a mathematical model to nationally representative data from the United States and showed that the difference in MM incidence can be explained by an increased incidence of MGUS among male and non-Hispanic Black populations. We did not find evidence showing differences in the rate of progression from MGUS to MM by either gender or race/ethnicity. Our results suggest that screening for MGUS among high-risk groups (e.g., non-Hispanic Black men) may hold promise as a strategy to reduce the burden and MM health disparities.

Multiple myeloma (MM) is a malignant transformation of plasma cells that caused an estimated 34,920 diagnoses and 12,410 deaths in the United States in 2021 alone[1]. MM is consistently preceded by a premalignant condition, monoclonal gammopathy of undetermined significance (MGUS)[2]. A prior cohort study of a predominately white population in Olmsted County, Minnesota suggested that MGUS progresses to MM at a rate of approximately 1% per year with the 20-year cumulative risk totaling 18%[3]. However, due to the absence of population-based screening and treatment recommendations for MGUS[4,5], most MGUS cases are detected incidentally[6], and, thus, our understanding of MGUS and MM are primarily informed by clinical studies. Further work is needed to understand the natural history of MM and how it varies across age, gender, and race/ethnicity.

One unresolved aspect of the natural history of MM concerns the observed disparities in MM incidence that exist by gender and race/ethnicity[7]. There is a greater burden of MM among men compared to women and non-Hispanic Black people compared to non-Hispanic white people. Additionally, both men and non-Hispanics Black people have a higher prevalence of MGUS[8–10] and develop MM earlier than women and non-Hispanic white people[11,12]. However, it remains unclear

whether the increased incidence of MM can be attributed to an increased incidence of MGUS, to an increased rate of progression from MGUS to MM, or a combination of both[7]. An improved understanding of the cause of these observed disparities can potentially guide future screening and treatment strategies aiming to reduce these disparities[13].

Prior studies have aimed to understand racial and ethnic disparities in MM incidence[10,11]. Landgren et al.[9] leveraged the National Health and Nutritional Examination Survey and found that high-risk features of MGUS were more common among non-Hispanic Black people as compared to non-Hispanic white people, suggesting an increased rate of progression from MGUS to MM among non-Hispanic Black people. However, this study lacked long-term outcomes on MM progression and was unable to confirm this finding. Using the Veterans Health Administration database, Landgren et al.[10] examined veterans diagnosed with MGUS during 1980–1996 and found no difference in rate of progression from MGUS to MM among non-Hispanic white people and non-Hispanic Black people. Therefore, they concluded that the higher MM incidence among non-Hispanic Black people could be explained by an increased incidence of MGUS. A more recent analysis

[1]Division of Public Health Sciences, Department of Surgery, Washington University School of Medicine, St. Louis, MO, USA. [2]Department of Electrical and Systems Engineering, Washington University in St. Louis, St. Louis, MO, USA. ✉e-mail: huber.j.h@wustl.edu

using the same database demonstrated progression to MM occurred at a younger age among non-Hispanic Black people as compared to non-Hispanic white people in patients diagnosed with MGUS, which may be attributed to the higher incidence of MGUS, the higher progression rate of MGUS to MM, or both in non-Hispanic Black people[13]. Nevertheless, reconciling the conclusions between these two studies is challenging, and, importantly, both studies relied upon study populations with clinically diagnosed MGUS. However, MGUS is primarily asymptomatic, so examining the progression of MM among patients with incidentally diagnosed MGUS excludes those with undiagnosed MGUS and therefore may provide an incomplete picture of the natural history of MM.

To address this, we leveraged two nationally representative databases from the United States on MGUS prevalence using data from the National Health and Nutritional Examination Survey (NHANES) and MM incidence using data from the Surveillance, Epidemiology, and End Results (SEER) program. Importantly, the NHANES tested all study participants for MGUS using serum protein electrophoresis irrespective of underlying comorbidities, providing more representative measures of MGUS prevalence than would be obtained using clinically diagnosed MGUS[14–16]. We constructed a compartmental model of the natural history of MM and fit this model using the aforementioned MGUS prevalence and MM incidence. We then used the fitted model to isolate the contributions of age, gender, and race/ethnicity to the observed disparities in MM incidence. Finally, we predicted how the preclinical dwell time, defined as the time from MGUS onset to MM onset, is likely to vary by age, gender, and race/ethnicity.

## Results
### Model fit
The fitted mathematical model of the natural history of MM shows that five independent MCMC chains were well-mixed (Fig. S2), and the Gelman-Rubin statistics for each parameter were 1.0 (Table S2),

providing support that we converged upon on the posterior distribution.

The fitted model captured the trends in MGUS prevalence and MM incidence across age, gender, and race/ethnicity (Fig. 1). Consistent with the data, the fitted model predicted higher MGUS prevalence and MM incidence among non-Hispanic Black people compared to non-Hispanic white people and among men compared to women. Furthermore, the fitted model reproduced the data with appropriate uncertainty. The 95% posterior prediction interval contained all but two data points with most data points falling near the posterior median prediction. Taken together, these results show that our fitted model can explain the patterns in the data.

### Contributions of age, gender, and race/ethnicity to the development of MGUS and progression to MM
Independent of gender and race/ethnicity, the rates of development of MGUS and progression to MM increased nonlinearly with age. The rate of MGUS development among healthy individuals monotonically increased with age (Fig. 2a, black line), nearly tripling from 0.0012 (95% Credible Interval (CI): 0.0099–0.0015) $yr^{-1}$ at age 60 to 0.0034 (0.0022–0.0049) $yr^{-1}$ by age 80. By comparison, the rate of progression to MM among MGUS-positive individuals monotonically increased up to age 71 (95% CI: 67–77) and then monotonically decreased. Modeling the rate of MM progression as a quadratic relationship with age allowed us to capture the declines in MM incidence seen in the 80–84 and 85+ age groups in the SEER data (Fig. 1, bottom row). Supplementary analyses found that including a quadratic term produced a better model fit to the data on the basis of DIC as compared to an alternative model in which the quadratic term was not included (see Supplementary Information for more details).

We estimated that gender and race/ethnicity modified the rates of development of MGUS and progression to MM with age. For healthy individuals, female gender reduced the rate of MGUS development by

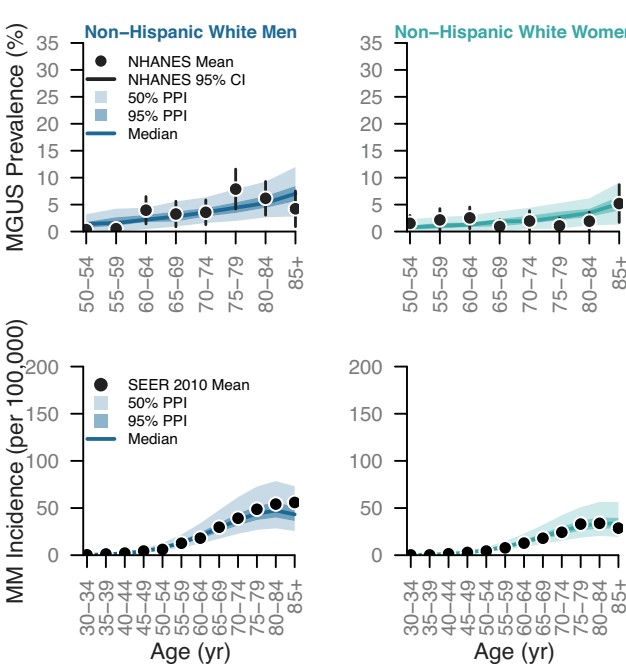
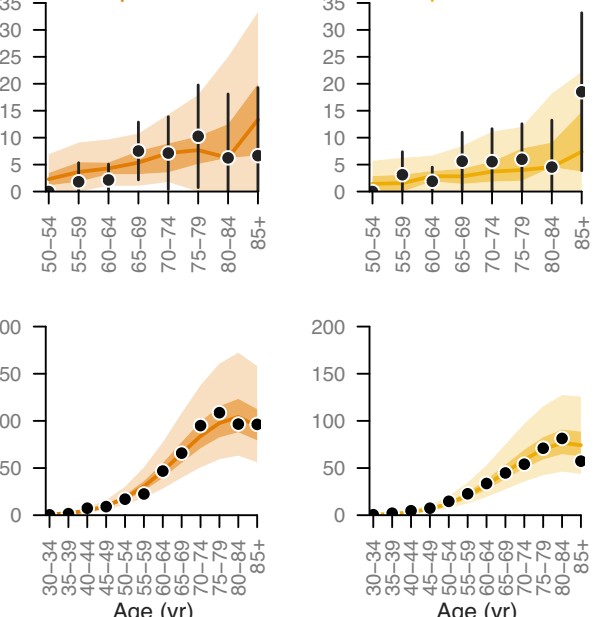

**Fig. 1 | Comparison of fitted model to SEER and NHANES data.** The model predictions for age-stratified MGUS prevalence and MM incidence are compared against the continuous NHANES 1999–2004 MGUS data (top row) and SEER 2010 MM incidence data (bottom row) for non-Hispanic white men (first column), non-Hispanic white women (second column), non-Hispanic Black men (third column), and non-Hispanic Black women (fourth column). Black points are the data used to fit the model, and the black vertical segments are the 95% confidence intervals on

the data. The 95% confidence intervals were calculated for each sub-population using the following sample sizes: $n = 1735$ non-Hispanic white men, $n = 1,703$ non-Hispanic white women, $n = 454$ non-Hispanic Black men, and $n = 463$ non-Hispanic Black women. The solid line is median posterior model prediction, the darker shaded area is the 50% posterior prediction interval (PPI), and the lighter shaded area is the 95% PPI.

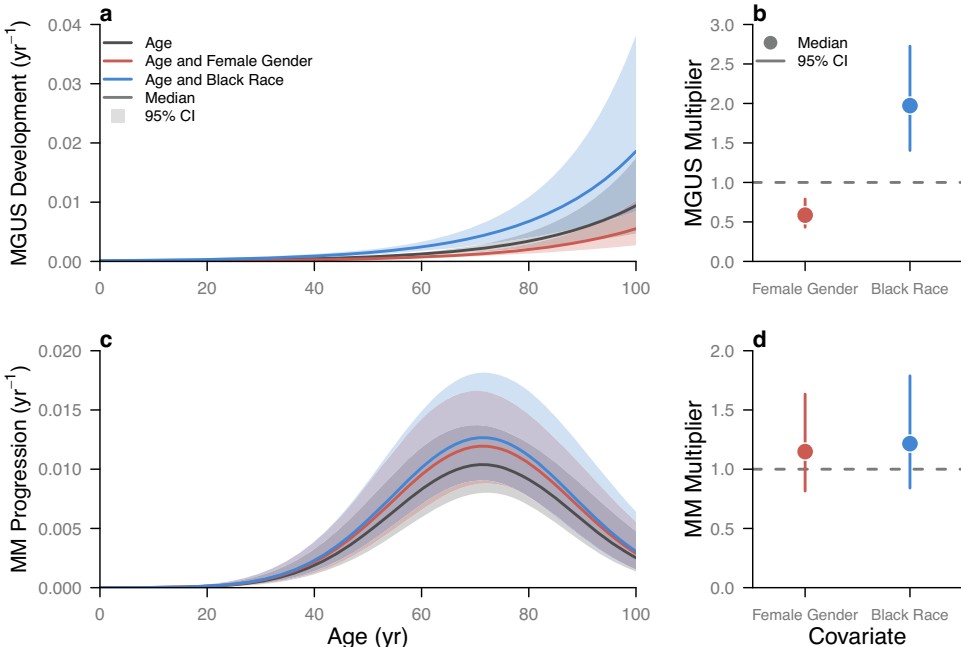

**Fig. 2 | Contributions of age, gender, and race/ethnicity to the development of MGUS and progression to MM. a** The isolated effects of age (black), age and female gender (red), and age and non-Hispanic Black race/ethnicity (blue) are shown for the rate of MGUS development (yr⁻¹). **b** The posterior estimate of the MGUS multiplier is shown for female gender (red) and non-Hispanic Black race/ethnicity (blue). **c** The isolated effect of age (black), age and female gender (red), and age and non-Hispanic Black race/ethnicity (blue) are shown for the rate of MM progression (yr⁻¹). **d** The posterior estimate of the MM multiplier is shown for

female gender (red) and non-Hispanic Black race/ethnicity (blue). In (**a**) and (**c**), the line is the median estimate, and the shaded region is the 95% credible interval. In (**b**) and **d**), the point is the median estimate, and the vertical line segment is the 95% credible interval. Each 95% credible interval is calculated from the $n = 50{,}010$ posterior samples. The horizontal dotted line is the reference multiplier of one. Values above the reference multiplier suggest that the covariate increases the rate of development/progression, whereas values below the reference multiplier suggest that the covariate decreases the rate of development/progression.

a multiplier of 0.59 (95% CI: 0.43–0.79) (Fig. 2a, red line; Fig. 2b). Similarly, non-Hispanic Black race/ethnicity increased the rate of MGUS development among healthy individuals by a multiplier of 2.0 (95% CI: 1.4–2.7) (Fig. 2a, blue line; Fig. 2b). That the 95% CI for both of these multipliers did not include 1.0 implies that female gender and non-Hispanic Black race/ethnicity significantly affect the rate of MGUS development across all age groups. By comparison, female gender and non-Hispanic Black race/ethnicity did not significantly affect the rate of progression to MM among MGUS-positive individuals (Fig. 2d). The multipliers on progression to MM were 1.1 (95% CI: 0.82–1.6) for female gender and 1.2 (0.84–1.8) for non-Hispanic Black race/ethnicity. Supplementary analyses in which we modified the relationships of the rates of MGUS development and MM progression with age or when we used SEER data from 2004 consistently found a significant effect of female gender and non-Hispanic Black race/ethnicity on MGUS development (see the Supplementary Information for more details).

### Duration of the preclinical dwell time

The preclinical dwell time, defined as the time from MGUS onset to MM onset, decreased with increasing age of MGUS onset (Fig. 3). For non-Hispanic white men, the expected preclinical dwell time was 16 (95% CI: 14–17) years at age 50, compared to 1.7 (95% CI: 1.6–1.8) years at age 90. As the age of onset increased, the variation in the expected preclinical dwell time similarly declined. This effect can be explained in part by the proportion of each cohort that survived to develop MM. For MGUS onset at age 50, 20% (95% CI: 16–26%) of non-Hispanic white men were expected to develop MM during their lifetime. By contrast, for MGUS onset at age 90, only 1.7% (95% CI: 1.1–2.6%) of non-Hispanic white men were expected to develop MM prior to death.

We estimated that the preclinical dwell time was affected by the gender and race/ethnicity of each cohort. Female gender was

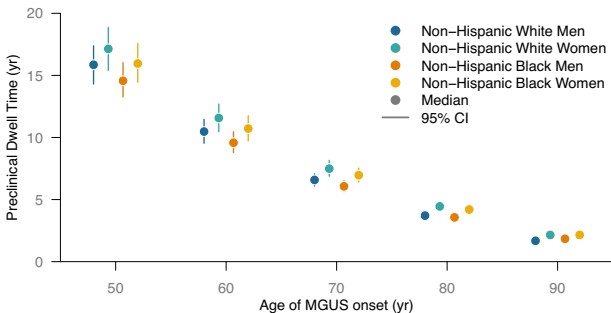

**Fig. 3 | Duration of the preclinical dwell time.** The posterior median (point) and the 95% credible interval (vertical line segment) is shown for the expected pre-clinical dwell time (i.e., the time from MGUS onset to MM onset) as a function of age for non-Hispanic white men (dark blue), non-Hispanic white women (light blue), non-Hispanic Black men (dark orange), and non-Hispanic Black women (light orange). Each 95% credible interval is calculated from $n = 50{,}010$ posterior samples.

associated with an increased dwell time across all ages of MGUS onset. For example, the expected dwell time at age 50 was 17 (95% CI: 15–19) years for non-Hispanic white women versus 16 (95% CI: 14–17) years for non-Hispanic white men and 16 (95% CI: 14–18) years for non-Hispanic Black women versus 15 (95% CI: 13–16) years for non-Hispanic Black men. Furthermore, independent of gender, non-Hispanic Black race/ethnicity was associated with a shorter preclinical dwell time. For instance, at age 70, the expected dwell times were 6.6 (95% CI: 6.1–7.1) years for non-Hispanic white men compared to 6.1 (95% CI: 5.6–6.5) years for non-Hispanic Black men and 7.5 (95% CI: 6.9–8.2) years for non-Hispanic white women compared to 7.0 (95% CI: 6.4–7.5) years for non-Hispanic Black women.

## Discussion

Obtaining a detailed understanding of the drivers of MM disparities is challenging, because MGUS, the preclinical state, is asymptomatic and most often detected incidentally[2]. Therefore, clinical studies, which examine progression to MM among patients with diagnosed MGUS exclude those with undiagnosed MGUS, likely biasing their analyses towards MGUS-positive patients with the greatest number of comorbidities[10,13]. By leveraging nationally representative data on MGUS prevalence[14–16], our study calibrated a discrete-time, multistate compartmental model of the natural history of MM that was able to uncover whether the higher incidence of MGUS, the progression rate of MGUS to MM, or both contributed to MM health disparities across age, gender, and race/ethnicity.

Prior studies have revealed that male gender and non-Hispanic Black race/ethnicity are risk factors for MGUS and MM, independent of other factors, such as age and socioeconomic status[8,9,17,18]. Our fitted model suggests that these disparities in MM incidence can be explained by an increased incidence of MGUS among healthy men and non-Hispanic Black people. Importantly, we found no statistically significant difference in the rate of progression from MGUS to MM, and these results were robust to multiple supplementary analyses that considered alternative models, including one in which there was no effect of MGUS on mortality, as well as alternative years of SEER MM incidence data. That the disparities in MM incidence can be explained by differential rate of development of MGUS suggests that strategies aiming to reduce disparities by gender and race/ethnicity should emphasize interventions that reduce the development of MGUS among high-risk groups. Although we identified an increased rate of development of MGUS amongst men and non-Hispanic Black people, our study cannot provide a mechanistic explanation for this phenomenon. It has been previously suggested that greater background plasma cell activity among non-Hispanic Black people may predispose them to developing MGUS and ultimately MM[19], and mutational signatures may be detectable in the early decades of life[20]. Alternatively, differences by race/ethnicity may be explained by socio-contextual factors[7] and differences in the distribution of known risk factors, such as obesity[13]. Future investigation that accounts for these and other hypotheses may eliminate the practice of essentializing race/ethnicity in cancer risk prediction models[21].

Independent of gender and race/ethnicity, we estimated the rates of developing MGUS and progressing from MGUS to MM increase nonlinearly with age. The rate of MGUS development monotonically increased with increasing age, suggesting that the observed increase in MGUS prevalence with age reflects a concomitant increase in MGUS incidence. This confirms a finding from a prior modeling study fitted to a predominately white cohort in Olmsted County, Minnesota and reveals that this finding is maintained across race/ethnicity as well as gender[22]. Furthermore, we estimated that the rate of progression from MGUS to MM peaked at approximately 71 years of age and subsequently declined, which mirrors the observed decline in MM incidence at higher age groups and may reflect a subset of older individuals with a more indolent presentation of MGUS and thus lower overall risk of progression to MM. Taken together, these results suggest that, if implemented, prevention strategies for MGUS may be cost-effective at all age groups, whereas prevention of MM among MGUS-positive individuals using pharmacological management, such as metformin[23] or aspirin[24], and non-pharmacological management, such as weight loss[13], may not be cost-effective beyond 71 years of age.

We estimated that the preclinical dwell time, defined as the time from MGUS onset to MM onset, declined nonlinearly with increasing age of MGUS onset. Because the preclinical dwell time requires that individuals with MGUS survive long enough to develop MGUS, we attribute this phenomenon to two competing effects: (1) the rate of progression from MGUS to MM and (2) the baseline mortality rate. The rate of progression from MGUS to MM increases nonlinearly up to age 71, resulting in a concomitant decline in the preclinical dwell time. Additionally, as individuals age, they are subject to a greater competing risk of mortality. This implies that, at higher ages of MGUS onset, those individuals that survive to progress to MM do so much more quickly, thereby shortening the average preclinical dwell time. This explains apparent differences in the preclinical dwell time between non-Hispanic Black people and non-Hispanic white people. Although we identified no statistically significant difference in the rate of progression from MGUS to MM across race/ethnicity, non-Hispanic Black people are subject to a higher mortality, resulting in shorter preclinical dwell times as compared to non-Hispanic white people.

Our study is subject to a number of limitations. First, we lacked data on smoldering multiple myeloma (SMM), an intermediate state between MGUS and MM, so we are unable to assess how progression to SMM varies across age, gender, and race/ethnicity[25]. Second, the data from NHANES and SEER are not collected from a single cohort. In order to utilize these distinct data sources within our modeling framework, we assumed that these data sources are nationally representative and thus reflect samples from the true MGUS prevalence and MM incidence within the U.S. population. Given the extensive geographic coverage and large samples sizes of these data sources[14–16,26], we believe the assumption is valid. However, the NHANES database is subject to non-response bias, which could affect our conclusions if the prevalence of MGUS was significantly different among those that responded as compared to those that did not respond. MGUS prevalence and MM incidence could also be underreported due to ascertainment bias. This would bias our estimates of the rates of MGUS development and progression from MGUS to MM downward and cause us to underestimate the preclinical dwell time. Additionally, extensions to our modeling framework could simulate multiple cohorts to account for period effects that likely shape MGUS prevalence and MM incidence. We considered only a single year of MGUS prevalence and MM incidence data. Our results were robust to the year of MM incidence data considered, but future work could assess whether increases in MM incidence over time can be attributed to an aging population or instead reflect an increase in MM risk over time. Finally, although we identified differences in MGUS development by gender and race/ethnicity, future work is needed to identify whether these differences can be explained by differences in the distribution of known risk factors, such as obesity[18], across gender and race/ethnicity.

This study found that disparities in MM incidence can be explained by an increased incidence of MGUS, not an increased rate of progression to MM, among healthy men and non-Hispanic Black people. Future studies are needed to identify whether these differences can be explained by differences in the distribution of known risk factors.

## Methods

Ethical approval was obtained from Washington University School of Medicine in St. Louis (IRB 202110041). All analyses were performed in accordance with relevant guidelines and regulations.

### Model overview

**Compartmental model.** To model the natural history of multiple myeloma[22], we constructed a discrete-time, multistate compartmental model consisting of four health states: healthy (H), monoclonal gammopathy of undetermined significance (MGUS), multiple myeloma (MM), and death (D) (Fig. 4).

For a given birth cohort of gender $s$ and race/ethnicity $r$, the proportion $P$ of the cohort in each of these states at a given age $a$ is defined by the following set of differential equations:

$$\frac{dP_H}{da} = -\lambda_{\text{MGUS}}(a,s,r)P_H - \mu_H(a,s,r)P_H, \qquad (1)$$

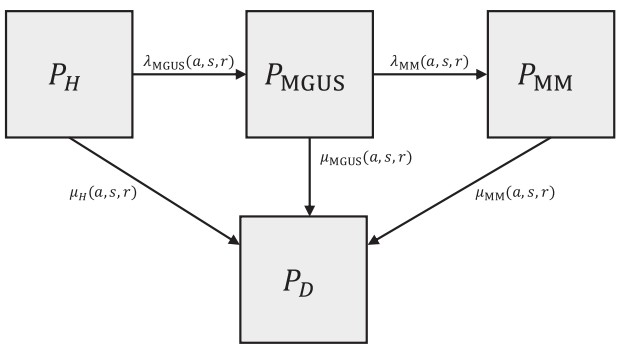

**Fig. 4 | Schematic of compartment model.** The schematic of the compartmental model of the natural history of multiple myeloma for a birth cohort is shown. Boxes represent the compartments, and arrows represent flows between compartments. $P_H$ is the proportion of the birth cohort that is healthy, $P_{MGUS}$ is the proportion of the birth cohort that has MGUS, $P_{MM}$ is the proportion of the birth cohort that has MM, and $P_D$ is the proportion of the both cohort that has died. $\lambda_{MGUS}(a,s,r)$ is the rate that a healthy individual of age a, gender s, and race/ethnicity r develops MGUS, and $\lambda_{MM}(a,s,r)$ is the rate that an individual of age a, gender s, and race/ethnicity r with MGUS develops MM. $\mu_H(a,s,r)$ is the mortality rate for a healthy individual of age a, gender s, and race/ethnicity r. $\mu_{MGUS}(a,s,r)$ is the mortality rate for an individual of age a, gender s, and race/ethnicity r with MGUS, and $\mu_{MM}(a,s,r)$ is the mortality rate for an individual of age a, gender s, and race/ethnicity r with MM.

$$\frac{dP_{MGUS}}{da} = \lambda_{MGUS}(a,s,r)P_H - \lambda_{MM}(a,s,r)P_{MGUS} - \mu_{MGUS}(a,s,r)P_{MGUS},$$
$$(2)$$

$$\frac{dP_{MM}}{da} = \lambda_{MM}(a,s,r)P_{MGUS} - \mu_{MM}(a,s,r)P_{MM},$$
$$(3)$$

$$\frac{dP_D}{da} = \mu_H(a,s,r)P_H + \mu_{MGUS}(a,s,r)P_{MGUS} + \mu_{MM}(a,s,r)P_{MM}.$$
$$(4)$$

In Eq. (1), $\lambda_{MGUS}(a,s,r)$ is the rate that a healthy individual of age $a$, gender $s$, and race/ethnicity $r$ develops MGUS, which we computed as

$$\lambda_{MGUS}(a,s,r) = e^{\gamma_{MGUS} + \beta_{MGUS,a}a + \beta_{MGUS,s} + \beta_{MGUS,r}}.$$
$$(5)$$

In Eq. (5), $\gamma_{MGUS}$ is an intercept term, such that $e^{\gamma_{MGUS}}$ denotes the baseline rate of developing MGUS independent of all other covariates. Additionally, $\beta_{MGUS,a}$, $\beta_{MGUS,s}$, and $\beta_{MGUS,r}$ are the coefficients that modulate the respective effects of age, gender, and race/ethnicity on the rate of MGUS development. In Eq. (2), $\lambda_{MM}(a,s,r)$ is the rate that an individual of age $a$, gender $s$, and race/ethnicity $r$ with MGUS develops MM and is calculated as $\lambda_{MM}(a,s,r) = e^{\lambda_{MM} + \beta_{MM,a}a + \beta_{MM,a2}a^2 + \beta_{MM,s} + \beta_{MM,r}}$. Finally, in Eqs. (1–4), $\mu_H(a,s,r)$ is the mortality rate for healthy individuals, $\mu_{MGUS}(a,s,r)$ is the mortality rate for individuals with MGUS, and $\mu_{MM}(a,s,r)$ is the mortality rate for individuals with MM. We followed Therneau et al.[22] by defining distinct mortality rates for individuals with MGUS and MM.

Equations (1–4) cannot be solved analytically and must be simulated forward in time numerically. Doing so yields $P_H(a,s,r)$, $P_{MGUS}(a,s,r)$, $P_{MM}(a,s,r)$, and $P_D(a,s,r)$, representing the proportion of the birth cohort of individuals of gender $s$ and race/ethnicity $r$ that occupies each state at age $a$.

**Prevalence and incidence.** The quantities $P_{MGUS}(a,s,r)$ and $P_{MM}(a,s,r)$ obtained from Eqs. (1–4) do not represent the respective prevalence of MGUS and MM, because the denominator includes individuals in the birth cohort that previously died. To calculate age-

stratified prevalence $p$ of MGUS and MM among individuals of gender $s$ and race/ethnicity $r$, we conditioned upon the proportion of the birth cohort that was alive at age $a$, such that

$$p_{MGUS}(a,s,r) = \frac{P_{MGUS}(a,s,r)}{1 - P_D(a,s,r)},$$
$$(6)$$

$$p_{MM}(a,s,r) = \frac{P_{MM}(a,s,r)}{1 - P_D(a,s,r)}.$$
$$(7)$$

Similarly, calculating age-stratified incidence $i$ of MGUS and MM among individuals of gender $s$ and race/ethnicity $r$ required that we condition upon the proportion of the birth cohort that was alive at age $a$. Therefore, we calculated $i_{MGUS}(a,s,r)$ and $i_{MM}(a,s,r)$ from Eqs. (1–7) as

$$i_{MGUS}(a,s,r) = \lambda_{MGUS}(a,s,r)p_H(a,s,r),$$
$$(8)$$

$$i_{MM}(a,s,r) = \lambda_{MM}(a,s,r)p_{MGUS}(a,s,r).$$
$$(9)$$

In Eq. (8), $p_H(a,s,r)$ is the prevalence of healthy individuals of age $a$, gender $s$, and race/ethnicity $r$, which we computed as $1 - p_{MGUS}(a,s,r) - p_{MM}(a,s,r)$ from Eqs. (6–7).

## Model fitting
**Data.** Using the continuous National Health and Nutritional Examination Surveys (NHANES), 1999–2004[14–16], we obtained empirical estimates of MGUS prevalence for 4,355 individuals 50 years of age and older stratified by age, gender, and race/ethnicity (Table 1). We aggregated the continuous NHANES into 5-year age bins in order to increase the number of MGUS-positive samples within each group. This data is the most current nationally representative survey on MGUS prevalence within the United States. Additionally, we obtained age-, gender-, and race/ethnicity-stratified estimates of MM incidence in 2010 from the Surveillance, Epidemiology, and End Results (SEER) Program[26]. MM incidence from 2010 was chosen specifically because the 6-year gap between the SEER and NHANES datasets approximates the gap between the 50–54 age group (i.e., the youngest age group in continuous NHANES for which MGUS data was available) and the 55–59 age group (i.e., the approximate age group for which MM incidence begins to substantially increase in SEER). To confirm that our conclusions were robust to the year of MM incidence data, we performed an alternative analysis in which used SEER data from 2004 (see sensitivity analysis in the Supplementary Information).

We obtained population estimates by age, gender, and race/ethnicity in 2010 from Centers for Disease Control (CDC) WONDER database[27]. For individuals 85 years of age or greater, population estimates were aggregated. To disaggregate population estimates for 85+ years, we assumed that the population distribution of individuals 85–99 years of age in 2010 was equivalent to the distribution in 2014, the first year in which CDC WONDER did not aggregate this age group[28].

**Table 1 | Characteristics of the data used for modeling fitting**

|  | Dataset | |
| --- | --- | --- |
|  | *Continuous NHANES* | *SEER* |
| **Measure** | MGUS Prevalence | MM Incidence |
| **Study Design** | Prevalence Survey | Observation |
| **Sample Size** | 4,355 individuals | 17 catchment areas |
| **Date** | 1999–2004 | 2010 |
| **Age Bins (years)** | 5 | 5 |
| **Stratification** | Gender, Race/Ethnicity | Gender, Race/Ethnicity |

For healthy individuals, we made use of age-, gender-, and race/ethnicity-stratified mortality rates in 2010 from the CDC Life Tables[29]. For MGUS-positive individuals, we followed Therneau et al.[22] and assumed that mortality in MGUS-positive individuals was 1.25 times greater than the baseline age- and race/ethnicity-specific mortality rate for men and 1.11 times greater than the baseline age- and race/ethnicity-specific mortality rate for women (see sensitivity analysis in the Supplementary Information). Finally, for individuals with MM, we estimated gender- and race/ethnicity-stratified all-cause mortality rates by fitting an exponential survival distribution to MM all-cause survival data provided by SEER (Fig. S1). Because these survival curves are derived from the cohort of all MM individuals within SEER, they provide an estimate of the mean all-cause mortality rate among MM individuals, irrespective of treatment characteristics.

**Likelihood function.** A crucial component of model fitting is the likelihood, which calculates the probability of the model given the data. In this study, the likelihood is equal to the product of the likelihoods for the two data types: (1) MGUS prevalence and (2) MM incidence. Because we adopt a fully Bayesian approach, we do not specify a weight for each data type during model fitting.

**MGUS prevalence.** We modeled the probability that $y_{a,s,r}$ individuals of age $a$, gender $s$, and race/ethnicity $r$ were MGUS-positive among $n_{a,s,r}$ surveyed individuals of age $a$, gender $s$, and race/ethnicity $r$ from NHANES as a binomial process, such that

$$\Pr\left(y_{a,s,r}|n_{a,s,r}, p_{\text{MGUS}}(a,s,r)\right) = p_{\text{MGUS}}(a,s,r)^{y_{a,s,r}}\left(1 - p_{\text{MGUS}}(a,s,r)\right)^{n_{a,s,r}-y_{a,s,r}}. \tag{10}$$

In Eq. (10), $p_{\text{MGUS}}(a,s,r)$ is the model-predicted prevalence of MGUS among individuals of age $a$, gender $s$, and race/ethnicity $r$ from Eqs. (1–4) and Eq. (6).

We aggregated the samples from the NHANES data into 5-year age bins in order to increase the number of MGUS-positive samples within each group. Accordingly, we calculated a weighted prevalence $\bar{p}_{\text{MGUS}}\left([a_1, a_2], s, r\right)$ between ages $a_1$ and $a_2$ as

$$\bar{p}_{\text{MGUS}}\left([a_1, a_2], s, r\right) = \frac{\sum_{a=a_1}^{a_2} N_{a,s,r} p_{\text{MGUS}}(a,s,r)}{\sum_{a=a_1}^{a_2} N_{a,s,r}}, \tag{11}$$

where $N_{a,s,r}$ is the size of the subpopulation of age $a$, gender $s$, and race/ethnicity $r$. For age-binned MGUS prevalence, the probability of observing $y_{[a_1, a_2], s, r}$ individuals between the ages of $a_1$ and $a_2$ among $n_{[a_1, a_2], s, r}$ total individuals surveyed is then calculated equivalently to Eq. (10) using the weighted predicted prevalence from Eq. (11). Therefore, the likelihood of the model given the NHANES data can be expressed as

$$\mathscr{L}\left(\vec{\mathbf{y}}, \vec{\mathbf{n}}, \vec{\mathbf{a}}_\mathbf{l}, \vec{\mathbf{a}}_\mathbf{u} | \vec{\boldsymbol{\theta}}\right)$$
$$= \prod_{i=1}^{n} \prod_{s \in \{M,F\}} \prod_{r \in \{\text{NHW}, \text{NHB}\}} \text{Binomial}\left(y_{[a_{l,i}, a_{u,i}], s, r} | n_{[a_{l,i}, a_{u,i}], s, r}, \bar{p}_{\text{MGUS}}\left([a_{l,i}, a_{u,i}], s, r\right)\right). \tag{12}$$

In Eq. (12), $\vec{\boldsymbol{\theta}}$ is the vector of parameters to be estimated, and $\vec{\mathbf{y}}$ and $\vec{\mathbf{n}}$ are the vectors of NHANES MGUS prevalence data where for each age bin the lower bound is defined by $\vec{\mathbf{a}}_\mathbf{l}$ and the upper bound is defined by $\vec{\mathbf{a}}_\mathbf{u}$.

**MM incidence.** Because MM incidence was reported as a continuous rate, we modeled the logarithm of MM incidence from SEER in individuals of age $a$, gender $s$, and race/ethnicity $r$ as a normal distribution with mean $\log(i_{\text{MM}}(a,s,r))$ and variance $\tau^2$ where $i_{\text{MM}}(a,s,r)$ is the model-predicted MM incidence among individuals of age $a$, gender $s$,

and race/ethnicity $r$ calculated from Eq. (9). We estimated $\tau^2$ as a parameter in our model and assumed that it did not depend upon age, gender, or race/ethnicity.

Similar to the MGUS prevalence data, MM incidence was binned by age. To accommodate this in our likelihood framework, we first calculated a weighted predicted incidence $\bar{i}([a_1, a_2], s, r)$ for individuals between ages $a_1$ and $a_2$ as

$$\bar{i}\left([a_1, a_2], s, r\right) = \frac{\sum_{a=a_1}^{a_2} N_{a,s,r} i_{\text{MM}}(a,s,r)}{\sum_{a=a_1}^{a_2} N_{a,s,r}}, \tag{13}$$

where $N_{a,s,r}$ is the population of individuals of age $a$, gender $s$, and $r$. The likelihood of the SEER MM incidence data can be expressed as

$$\mathscr{L}\left(\vec{\mathbf{x}}, \vec{\mathbf{a}}_\mathbf{l}, \vec{\mathbf{a}}_\mathbf{u} | \vec{\boldsymbol{\theta}}\right) = \prod_{i=1}^{n} \prod_{s \in \{M,F\}} \prod_{r \in \{\text{NHW}, \text{NHB}\}} \text{Normal}\left(\log\left(x_{[a_{l,i}, a_{u,i}], s, r}\right) | \bar{i}\left([a_{l,i}, a_{u,i}], s, r\right), \tau^2\right). \tag{14}$$

In Eq. (14), $\vec{\boldsymbol{\theta}}$ is the vector of estimated parameters, and $\vec{\mathbf{x}}$ is the vector of SEER MM incidence where for each age bin the lower bound is defined by $\vec{\mathbf{a}}_\mathbf{l}$ and the upper bound is defined by $\vec{\mathbf{a}}_\mathbf{u}$.

**Priors.** We assumed uniform prior distributions for all model parameters (Table 2). The choice of the upper and lower bound for the prior distribution for each parameter was informed by the plausible ranges that yielded real model output. In general, we specified wide upper and lower bounds to allow the inference algorithm the flexibility to explore the parameter space. However, we restricted the prior distribution of $\beta_{\text{MGUS},a}$ to [0,1]. The lower bound of this distribution was chosen because MGUS prevalence has been observed to increase with age[3]. Nevertheless, a sensitivity analysis was performed to evaluate how the choice of prior distribution affected the parameter inferences.

**Markov chain Monte Carlo.** We estimated the parameters of our model from the NHANES and SEER data using a Markov chain Monte Carlo (MCMC) algorithm[30]. We ran the MCMC for 1,000,000 samples, applied a burn-in of 500,000 samples, and thinned every 50 samples to reduce autocorrelation, thereby obtaining a posterior distribution of 10,000 samples. We assessed convergence by running five chains in parallel and computing the Gelman-Rubin statistic for each parameter, where values less than 1.1 provide statistical support for convergence[31]. Converged chains were pooled to yield a final posterior distribution of 50,000 samples.

## Analyses

After fitting the model and comparing the model predictions to the NHANES and SEER data, we used the fitted model to explore the epidemiology of MGUS and MM by age, gender, and race/ethnicity. First, we analyzed the estimated model parameters to isolate the contributions of age, gender, and race/ethnicity to the rates of progression

**Table 2 | Parameter definitions and prior distributions**

| Parameter | Definition | Prior |
|---|---|---|
| $\gamma_{MGUS}$ | Intercept for rate of MGUS development | [–20, 0] |
| $\beta_{MGUS,a}$ | Age coefficient for rate of MGUS development | [0, 1] |
| $\beta_{MGUS,s}$ | Gender coefficient for rate of MGUS development | [–15,5] |
| $\beta_{MGUS,r}$ | Race/ethnicity coefficient for rate of MGUS development | [–15, 5] |
| $\gamma_{MM}$ | Intercept for rate of MM development | [–20, 0] |
| $\beta_{MM,a}$ | Age coefficient for rate of MM development | [–15, 1] |
| $\beta_{MM,a^2}$ | Quadratic age coefficient for rate of MM development | [–15, 1] |
| $\beta_{MM,s}$ | Gender coefficient for rate of MM development | [–15, 5] |
| $\beta_{MM,r}$ | Race/ethnicity coefficient for rate of MM development | [–15, 5] |
| $\tau^2$ | Variance of MM incidence | [0, 100] |

from healthy to MGUS and from MGUS to MM. Next, we computed the expected duration of the preclinical dwell time (i.e., time from MGUS onset to MM onset) by age, gender, and race/ethnicity. The preclinical dwell time depends upon the rate of progression from MGUS to MM and the competing baseline mortality rate, both of which depend upon age, gender, and race/ethnicity. This relationship between the preclinical dwell time and the baseline mortality rate exists because individuals with MGUS must survive sufficiently long to progress to MM.

## Reporting summary

Further information on research design is available in the Nature Portfolio Reporting Summary linked to this article.

## Data availability

Databases used: National Health and Nutritional Examination Survey (NHANES) 1999–2004 (https://www.cdc.gov/nchs/nhanes/index.htm); CDC Life Tables (https://www.cdc.gov/nchs/products/life_tables.htm); CDC Wonder Population Database (https://wonder.cdc.gov/Bridged-Race-v2020.HTML); SEER*STAT Cancer Incidence (https://seer.cancer.gov/seerstat/); All data used in this analysis can be found at https://zenodo.org/record/8244914[32].

## Code availability

All code to reproduce the analyses can be found at https://zenodo.org/record/8244914[32]. Analyses were conducted in R using the BayesianTools and pomp R packages.

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

## Acknowledgements

This work was funded by the National Institute of Health's National Cancer Institute Grant Number R01CA453475 (S-H.C.) and as part of the Cancer Intervention and Surveillance Modeling Network (CISNET), Grant Number U01CA265735 (S-H.C. and G.C.). The funder had no role in the design of the study; the collection, analysis, and interpretation of the data; the writing of the manuscript; and the decision to submit the manuscript for publication. The article's contents are solely the responsibility of the authors and do not necessarily represent the official views of the National Cancer Institute. The authors thank the Washington University Information Technology's Research Infrastructure Services for computing support.

## Author contributions

J.H.H, G.C., and S-H.C. conceived of the study. S-H.C. and G.C. obtained the funding. M.W. and M.J. collected the data. J.H.H. and S-H.C. developed the methods and conducted the analyses. J.H.H., M.J., M.W., Y.S., and S-H.C. analyzed the results. J.H. and S-H.C. wrote the initial draft. All authors provided input into the manuscript and approved of its final form.

## Competing interests

The authors declare no competing interests.
