## [Peer Review File · Nature Communications]

Disentangling age, gender, and racial/ethnic disparities in multiple myeloma burden: a modeling studyReviewers' comments:

Reviewer #1 (Remarks to the Author):

Huber et al. generate a mathematical model of the natural history of multiple myeloma, using data from Mhanes and SEER. Using this model they demonstrate:

- a) the rate of MGUS development was higher in men and in non-Hispanic blacks
- b) the rates of progression of MGUS to MM increased non-linearly with age with no difference between sex/ethnicity
- c) the pre-clinical dwell time decreased with the age of MGUS onset

The scientific question is of interest and model clearly described.

I have a number of concerns:

Major comments:

1. Many of the findings have been described in prior work.

a) A study in NHANES (from which data has been used to generate the model in this manuscript) (<https://www.nature.com/articles/leu201434>), demonstrates MGUS is more common in black individual. This has also been demonstrated in two cohort studies:

(<https://ashpublications.org/bloodadvances/article/6/12/3746/484453/Prevalence-of-heavy-chain-MGUS-by-race-and-family> and <https://ashpublications.org/blood/article/107/3/904/22154/Risk-of-monoclonal-gammopathy-of-undetermined>)

b) The preclinical dwell time (defined as progression from MGUS to MM) was shorter for black individuals. Conflicting evidence exists in the literature but an increased rate of progression has been described in black individuals (<https://www.nature.com/articles/leu201434>).

c) The rate of MGUS development monotonically increased with increasing age as described here: [https://linkinghub.elsevier.com/retrieve/pii/S0025-6196\(12\)00634-9](https://linkinghub.elsevier.com/retrieve/pii/S0025-6196(12)00634-9)

2. The authors state the pre-clinical dwell time is shorter for black individuals e.g. "Furthermore, independent of sex, black race was associated with a shorter preclinical dwell time. For instance, at age 70, the expected dwell times were 6.6 (95% CI: 6.1 – 7.1) years for non-Hispanic white males compared to 6.1 (95% CI: 5.6 – 6.5) years for non Hispanic black males". They also state that the rate of MM development (for a given time period) in MGUS individuals were not affected by race. Unless I have misunderstood is there not a contradiction in these statements?

3. The NHANES and SEER have biases including non-responder biases (e.g. https://www.cdc.gov/nchs/data/bsc/bscpres_fakhouri_january_2018.pdf). The authors should discuss this, how this might impact on their model and how it would affect their assertion that their model can address inconsistencies in the literature.

4. Whilst simulation studies were used to assess the validity of their model. Independent validation of their model would bring strength to this manuscript.

5. There are a number of important references missing - discussion would benefit the manuscript:

1) <https://www.nature.com/articles/leu201434>

2) <https://www.nature.com/articles/s41467-020-15740-9>

3) <https://ashpublications.org/bloodadvances/article/6/12/3746/484453/Prevalence-of-heavy-chain-MGUS-by-race-and-family>

Reviewer #2 (Remarks to the Author):

Huber et al. conducted an important study which found that the increased incidence of MGUS can explain disparities in MM. Therefore, screening and prevention of MGUS among high-risk groups hold promise as strategies to reduce the burden of MM. The paper is well written with robust statistical analysis conducted. I have a few comments for further improvement of the paper.

- 1) Data used were relatively old: 1999-2004 for NHANES and 2010 for SEER. It would be better if updated data (2020) can be used for the modelling.
- 2) The authors should include a Table explaining the different characteristics between NHANES and SEER.
- 3) The assumptions made should be highlighted in the abstract and discussion of the paper.
- 4) Asian population should also be included in the analysis.
- 5) Is it possible to adjust for drug use in the analysis?
- 6) Obesity is another risk factor to be considered in the analysis.
- 7) Socioeconomic status should also be considered to explain the disparities observed.
- 8) Any explanations for the progression rate from MGUS to MM peaked at 71 years of age?

Reviewer #3 (Remarks to the Author):

This study describes a mathematical model of multiple myeloma natural history, particularly the transition from MGUS to MM. The authors use data from NHANES on the age-specific prevalence of MGUS and from SEER on the age-specific incidence of MM. They define a multi-state model and formulate the dynamics of the various transitions as a set of differential equations parameterized by the transition rates. They then estimate the transition rates by formulating a likelihood function for the incidence and prevalence data and applying a Bayesian MCMC algorithm. The model is fit separately to population subgroups defined by sex and race to infer whether the data are consistent with similar rates of MGUS onset or transition from MGUS to MM across age- and race groups.

This is very informative and could offer direction for interventions to prevent MM in the different subgroups. I have some key concerns however about the methods and results.

First, regarding the likelihood development, the use of a normal distribution for the incidence piece does not really comport with the count data nature of incidence figures for which the mean and variance should be related e.g. via a Poisson or other count data type model and also informed by the sample size.

Second, the incidence information is on a different scale than the prevalence information, and it is unclear, when the two parts of the likelihood are multiplied, how this is weighted relative to the prevalence? In other words, in the model fitting, is the fit prioritizing prevalence or incidence? One thing that strikes me is that the prevalence of MGUS by age actually contains information on both the incidence of MGUS and the transition rate to MM if the rate of other-cause death among people with MGUS is known (indeed it is assumed known and provided as an input to the model). So, the model could in principle be fit just to the prevalence data and then validated against the incidence data. I wonder if the investigators have considered this? At any rate, some discussion of the validity and implications of the product formulation of the likelihood seems important including some insight into the relative contributions of the incidence and prevalence data.

Finally, I understand the explanation for the pattern of MGUS duration by age in Figure 4, but I don't understand why the authors could not produce a version of this figure in the absence of other-cause death, which would be a much cleaner comparison of natural history across ages. Can this be added alongside or as an additional figure?

The Discussion does not address the limitation of the model structure in that it is essentially age-group-specific and does not lend itself to aging from one age group into another (MM incidence in

a given age group depends not only on MGUS prevalence in that group but on MGUS prevalence in prior groups). It would be of value to understand this and to see some discussion of alternative modeling approaches that have been used for cancer natural history modeling e.g. continuous-time multi-state models.

REVIEWER 1

Comment 1.1

Huber et al. generate a mathematical model of the natural history of multiple myeloma, using data from MHANES and SEER. Using this model they demonstrate:

- a) the rate of MGUS development was higher in men and in non-Hispanic blacks*
 - b) the rates of progression of MGUS to MM increased non-linearly with age with no difference between sex/ethnicity*
 - c) the pre-clinical dwell time decreased with the age of MGUS onset*
- The scientific question is of interest and model clearly described.
I have a number of concerns.*

Response 1.1

We thank the reviewer for this summary of our work. We believe that the recommendations made by the reviewer have helped us to clarify how our work builds upon prior studies and have strengthened our manuscript.

Comment 1.2

Many of the findings have been described in prior work.

- a) A study in NHANES (from which data has been used to generate the model in this manuscript) (<https://www.nature.com/articles/leu201434>), demonstrates MGUS is more common in black individual. This has also been demonstrated in two cohort studies: (<https://ashpublications.org/bloodadvances/article/6/12/3746/484453/Prevalence-of-heavy-chain-MGUS-by-race-and-family> and <https://ashpublications.org/blood/article/107/3/904/22154/Risk-of-monoclonal-gammopathy-of-undetermined>)*
- b) The preclinical dwell time (defined as progression from MGUS to MM) was shorter for black individuals. Conflicting evidence exists in the literature but an increased rate of progression has been described in black individuals (<https://www.nature.com/articles/leu201434>).*
- c) The rate of MGUS development monotonically increased with increasing age as described here: [https://linkinghub.elsevier.com/retrieve/pii/S0025-6196\(12\)00634-9](https://linkinghub.elsevier.com/retrieve/pii/S0025-6196(12)00634-9)*

Response 1.2

We thank the reviewer for identifying studies relevant to our work. We agree with the reviewer that racial disparities in MGUS and multiple myeloma is an area of intense focus. However, we believe that our study fills a gap in the literature not addressed by the referenced studies above. Specifically, we adopted a mechanistic modeling approach to link two nationally representative databases on MGUS prevalence and MM incidence to better understand the contribution to age, sex, and race to observed disparities in MM incidence. We agree with the reviewer that Landgren *et al.* (2014) and subsequent studies established that MGUS is more prevalent among non-Hispanic black populations. However, our study answers three unique questions, which have not been able to be answered in the past studies – 1) whether this increased MGUS prevalence and/or the increased transformation of MGUS to MM was contributing to racial difference in MM incidence; 2) how long the preclinical dwell time is as a function of age, sex, and race; and 3) factors contributing to the preclinical dwell time.

Finally, we agree with the reviewer that Therneau *et al.* (2012) demonstrated that the rate of MGUS development increased monotonically with age in a predominately white cohort. Nonetheless, our study further demonstrates that that this result holds true for non-Hispanic black populations. Additionally, we show that the rate of MM progression has a nonlinear relationship with age and importantly peaks around age 70 and that individuals greater than 70 years of age have a lower rate of MM progression, even when adjusting for all-cause mortality, which fills the gap in the literature and warrants further investigation.

We appreciate the reviewer raising all of these points as we believe that it has helped to better communicate the gap in the literature that our study addresses. We reference these studies now throughout the manuscript to better highlight the prior literature in this area.

Comment 1.3

The authors state the pre-clinical dwell time is shorter for black individuals e.g. "Furthermore, independent of sex, black race was associated with a shorter preclinical dwell time. For instance, at age 70, the expected dwell times were 6.6 (95% CI: 6.1 – 7.1) years for non-Hispanic white males compared to 6.1 (95% CI: 5.6 – 6.5) years for non Hispanic black males". They also state that the rate of MM development (for a given time period) in MGUS individuals were not affected by race. Unless I have misunderstood is there not a contradiction in these statements?

Response 1.3

We apologize for the confusion. This appears to be a contradiction, because pre-clinical dwell time is determined by both 1) the rate of MM development; and 2) the mortality rate. Although we identified no statistically significant difference in the rate of MM development across race, non-Hispanic blacks are subject to a higher mortality rate, thereby shortening the pre-clinical dwell time. We clarify this in the Discussion on Lines 417-421:

"This explains apparent differences in the preclinical dwell time between non-Hispanic blacks and whites. Although we identified no statistically significant difference in the rate of progression from MGUS to MM across race/ethnicity, non-Hispanic blacks are subject to a higher mortality, resulting in shorter preclinical dwell times as compared to non-Hispanic whites."

Comment 1.4

The NHANES and SEER have biases including non-responder biases (e.g. https://www.cdc.gov/nchs/data/bsc/bscpres_fakhouri_january_2018.pdf). The authors should discuss this, how this might impact on their model and how it would affect their assertion that their model can address inconsistencies in the literature.

Response 1.4

We thank the reviewer for raising this. Ultimately, this bias in the NHANES is known, and we are unable to comment on the direction it would have on our conclusions, given that we do not know the prevalence of MGUS and among non-respondents as compared to respondents. We do not believe that SEER data is subject to the same bias, because SEER data are submitted by cancer registries. We have acknowledged this limitation in the Discussion on Lines 429-431:

“However, the NHANES database is subject to non-response bias, which could affect our conclusions if the prevalence of MGUS was significantly different among those that responded as compared to those that did not respond.”

Comment 1.5

Whilst simulation studies were used to assess the validity of their model. Independent validation of their model would bring strength to this manuscript.

Response 1.5

We thank the reviewer for this suggestion. We compared model predictions for MGUS prevalence to those reported by Kyle *et al.* (2006) in Olmsted County, Minnesota. Additionally, we compared our model predictions for the lifetime risk of developing MM to estimates published by the American Cancer Society. A description of the methods taken be found in the supplement on lines 534-544:

Independent Validation

We performed an independent validation of our fitted model by comparing our model predictions to data sources that were not used to fit the model. Specifically, we compared predicted MGUS prevalence by age and sex from our model to predicted MGUS prevalence by age and sex in Olmsted County, Minnesota between 1995 and 2001³. Because 97.3% of the Olmsted County cohort identified as white, we made use of model predictions for non-Hispanic white males and females. Next, we compared our model’s prediction for lifetime risk of developing MM to estimates reported by the American Cancer Society’s Cancer Statistics from 2023³⁰. Estimates were only available by sex, so we compared our model predictions for non-Hispanic whites, non-Hispanic blacks, and a composite of non-Hispanic whites and non-Hispanic blacks weighted by population size.

The results of this independent validation can be found in the supplement on lines 665-693:

Independent Validation

We compared model estimates of MGUS prevalence for non-Hispanic white males and females to estimates from Olmsted County, Minnesota between 1995-2001³. Although the data used for validation reflects a single cohort from a separate time period in one geographical location, there is reasonably good agreement between the model predictions and the validation data with respect to the magnitude of MGUS prevalence and its relationship with age (Fig. S6). A previous study comparing NHANES 1999-2003 to the Olmsted County cohort noted higher MGUS prevalence in Olmsted County, suggesting that the higher MGUS prevalence in Olmsted County may reflect geographical variation⁹.

Figure S6. Independent validation of MGUS prevalence for non-Hispanic white males and females. Model predictions of MGUS prevalence for non-Hispanic white (A) males and (B) females are shown as a function of age and compared to estimates from Kyle et al.³ in Olmsted County, Minnesota. The dashed black line is the estimate from Kyle et al.³. The solid line is median posterior model prediction, the darker shaded area is the 50% credible interval (CI), and the lighter shaded area is the 95% CI.

We additionally compared our model estimates of lifetime risk of developing MM to estimates published by the American Cancer Society³⁴. Reported lifetime risk of developing MM for males is 0.9%. Our model predictions for the lifetime risk of developing MM were 0.77% (95% CI: 0.68 – 0.87%) for non-Hispanic white males and 1.42% (95% CI: 1.25 – 1.59%) for non-Hispanic black males. Weighting each race/ethnicity by population size, we obtained a lifetime risk of developing MM for males of 0.88% (95% CI: 0.77 – 0.99%), comparable to the estimate reported by the American Cancer Society. For females, reported lifetime risk of developing MM is 0.7%. Our model predictions for the lifetime risk of developing MM were 0.66% (95% CI: 0.57 – 0.75%) for non-Hispanic white females and 1.33% (95% CI: 1.18 – 1.48%) for non-Hispanic black females. Weighting each race/ethnicity by population size, we obtained a lifetime risk of developing MM for females of 0.77% (95% CI: 0.68 – 0.87%), comparable to the estimate reported by the American Cancer Society.

Comment 1.6

There are a number of important references missing - discussion would benefit the manuscript:

- 1) <https://www.nature.com/articles/leu201434>
- 2) <https://www.nature.com/articles/s41467-020-15740-9>
- 3) <https://ashpublications.org/bloodadvances/article/6/12/3746/484453/Prevalence-of-heavy-chain-MGUS-by-race-and-family>

Response 1.6

We thank the reviewer for these references. We have cited them in the discussion.

REVIEWER 2

Comment 2.1

Huber et al. conducted an important study which found that the increased incidence of MGUS can explain disparities in MM. Therefore, screening and prevention of MGUS among high-risk groups hold promise as strategies to reduce the burden of MM. The paper is well written with robust statistical analysis conducted. I have a few comments for further improvement of the paper.

Response 2.1

We thank the reviewer for this summary and assessment of our work. We hope that the revised manuscript addresses the concerns raised by the reviewer.

Comment 2.2

Data used were relatively old: 1999-2004 for NHANES and 2010 for SEER. It would be better if updated data (2020) can be used for the modelling.

Response 2.2

We agree with the reviewer that we would prefer to have more updated data. The MGUS prevalence survey from 1999-2004 is the most recent nationally representative MGUS survey. We choose the SEER data from 2010 to be approximately comparable in the age groups considered in the NHANES data from 1999-2004. We clarify that the NHANES data is the most current nationally representative data source available on Lines 164-165:

“This data is the most current nationally representative survey on MGUS prevalence within the United States.”

Comment 2.3

The authors should include a Table explaining the different characteristics between NHANES and SEER.

Response 2.3

We thank the reviewer for this suggestion. We have included a table in the Methods on Lines 191-192:

Table 2. Characteristics of the Data used for Modeling Fitting

Dataset	Measure	Study Design	Sample Size	Date	Age Bins	Stratification
Continuous NHANES	MGUS Prevalence	Prevalence Survey	4,355 individuals	1999-2004	5 years	Sex, Race
SEER	MM Incidence	Observational	17 catchment areas	2010	5 years	Sex, Race

Comment 2.4

The assumptions made should be highlighted in the abstract and discussion of the paper.

Response 2.4

We thank the reviewer for this suggestion. Due to the word limit of the abstract, we are limited in the assumptions that we can list there. However, we have specified the model structure on Line 20:

“We constructed a discrete time, multi-state compartmental model of the natural history of MM.”

Additionally, we specify the model structure in the Discussion on Lines 372-376:

By leveraging nationally representative data on MGUS prevalence¹⁴⁻¹⁶, our study calibrated a discrete time, multi-state compartmental model of the natural history of MM that was able to uncover whether the higher incidence of MGUS, the progression rate of MGUS to MM, or both contributed to MM health disparities across age, sex, and race/ethnicity.

Finally, we expanded in the Discussion on Lines 380-384 on the supplementary analyses that were performed to clarify some of the assumptions that were made and how robust our analyses were to these assumptions:

“Importantly, we found no statistically significant difference in the rate of progression from MGUS to MM, and these results were robust to multiple supplementary analyses that considered alternative models, including one in which there was no effect of MGUS on mortality, as well as alternative years of SEER MM incidence data.”

Comment 2.5

Asian population should also be included in the analysis.

Response 2.5

The NHANES MGUS prevalence survey did not identify Asian as a demographic population. There was a category for “other non-Hispanic white,” but this was not specific to the Asian population alone. As such, we limited our analysis to non-Hispanic white and non-Hispanic black populations.

Comment 2.6

Is it possible to adjust for drug use in the analysis?

Response 2.6

The reviewer raises an interesting point regarding drug use and treatment for multiple myeloma. In our analysis, we made use of all-cause survival data for individuals diagnosed with multiple myeloma from SEER. Because this considers a cohort of MM patients, irrespective of treatment modality or lack thereof, we obtained a mean all-cause mortality rate for the subset of the population diagnosed with multiple myeloma. We felt that this was appropriate given that our analysis was not focused on estimating the effects of various treatment regimens and was considered at a population scale. We clarify this in the text on Lines 185-189:

“Finally, for individuals with MM, we estimated sex- and race/ethnicity-stratified all-cause mortality rates by fitting an exponential survival distribution to MM all-cause survival data provided by SEER (Fig. S1). Because these survival curves are derived from the cohort of all MM individuals within SEER, they provide an estimate of the mean all-cause mortality rate among MM individuals, irrespective of treatment characteristics.”

Comment 2.7

Obesity is another risk factor to be considered in the analysis.

Response 2.7

We thank the reviewer for this suggestion. Obesity is a known risk factor for progression to MM and likely promotes development of MGUS as well, as also demonstrated by a previous study conducted by our team. Because the distributions of BMI, race/ethnicity, and sex are highly correlated, it is likely that some of the estimated effects of sex and race/ethnicity on MGUS development can be explained by obesity. We comment on this in the Discussion on Lines 391-395:

“Alternatively, differences by race/ethnicity may be explained by socio-contextual factors⁷ and differences in the distribution of known risk factors, such as obesity¹³. Future investigation that accounts for these and other hypotheses may eliminate the practice of essentializing race/ethnicity in cancer risk prediction models²⁸.”

BMI was available for NHANES data, but not SEER incidence. Accordingly, we chose to not estimate the effect of BMI in this study, but instead comment on its possible effect in the Discussion. Incorporating the effect of BMI on MGUS development and MM progression is an active area of work for this group.

Comment 2.8

Socioeconomic status should also be considered to explain the disparities observed.

Response 2.8

We thank the reviewer for this suggestion. We include this as a possible explanation for the patterns by race/ethnicity and sex on Lines 391-395:

“Alternatively, differences by race/ethnicity may be explained by socio-contextual factors⁷ and differences in the distribution of known risk factors, such as obesity¹³. Future investigation that accounts for these and other hypotheses may eliminate the practice of essentializing race/ethnicity in cancer risk prediction models²⁸.”

Comment 2.9

Any explanations for the progression rate from MGUS to MM peaked at 71 years of age?

Response 2.9

The peak in the progression rate from MGUS to MM at 71 years of age reflects the decline in MM incidence seen in later age groups. Given that we accounted for the competing risk of mortality through age-, race- and sex-stratified all-cause mortality rates, it is possible that the decline in the estimated progression rate at higher age groups may be capturing MGUS-positive individuals with characteristics of lower risk of transformation. That is to say, conditional on surviving up beyond age 71 without transforming to MM, you may have MGUS with features that are less likely to transform to MM. We comment on this possibility on Lines 402-405:

“Furthermore, we estimated that the rate of progression from MGUS to MM peaked at approximately 71 years of age and subsequently declined, which mirrors the observed decline in MM incidence at higher age groups and may reflect a subset of older individuals with a more indolent presentation of MGUS and thus lower overall risk of progression to MM.”

REVIEWER 3

Comment 3.1

This study describes a mathematical model of multiple myeloma natural history, particularly the transition from MGUS to MM. The authors use data from NHANES on the age-specific prevalence of MGUS and from SEER on the age-specific incidence of MM. They define a multi-state model and formulate the dynamics of the various transitions as a set of differential equations parameterized by the transition rates. They then estimate the transition rates by formulating a likelihood function for the incidence and prevalence data and applying a Bayesian MCMC algorithm. The model is fit separately to population subgroups defined by sex and race to infer whether the data are consistent with similar rates of MGUS onset or transition from MGUS to MM across age- and race groups.

This is very informative and could offer direction for interventions to prevent MM in the different subgroups. I have some key concerns however about the methods and results.

Response 3.1

We thank the reviewer for this summary and assessment of our work. We hope that the revised manuscript addresses the concerns raised by the reviewer.

Comment 3.2

First, regarding the likelihood development, the use of a normal distribution for the incidence piece does not really comport with the count data nature of incidence figures for which the mean and variance should be related e.g. via a Poisson or other count data type model and also informed by the sample size.

Response 3.2

We apologize that this was not clear. The incidence estimates reported by SEER are rates, not count data. Since they are continuous, not discrete in nature, we did not feel that a count data type model was appropriate. We clarify that the data is a continuous rate on Lines 228-231:

“Because MM incidence was reported as a continuous rate, we modeled the logarithm of MM incidence from SEER in individuals of age a , sex s , and race r as a normal distribution with mean $\log(i_{MM}(a, s, r))$ and variance τ^2 where $i_{MM}(a, s, r)$ is the model-predicted MM incidence among individuals of age a , sex s , and race r calculated from eq. (9).”

Comment 3.3

Second, the incidence information is on a different scale than the prevalence information, and it is unclear, when the two parts of the likelihood are multiplied, how this is weighted relative to the prevalence? In other words, in the model fitting, is the fit prioritizing prevalence or incidence? One thing that strikes me is that the prevalence of MGUS by age actually contains

information on both the incidence of MGUS and the transition rate to MM if the rate of other-cause death among people with MGUS is known (indeed it is assumed known and provided as an input to the model). So, the model could in principle be fit just to the prevalence data and then validated against the incidence data. I wonder if the investigators have considered this? At any rate, some discussion of the validity and implications of the product formulation of the likelihood seems important including some insight into the relative contributions of the incidence and prevalence data.

Response 3.3

We thank the reviewer for raising this point. We adopted a fully Bayesian approach, so we did not weight the prevalence and incidence data. We feel that this is the appropriate approach because we are not specifying beforehand which data that we feel are more “important,” per se. We note that the data is not weighted in the likelihood on Lines 196-197:

“Because we adopt a fully Bayesian approach, we do not specify a weight for each data type during model fitting.”

As the reviewer notes, depending upon the likelihood, the model could be fit more to one data type at the expense of the other data type. However, as can be observed in Figure 2, the fitted model is able to capture the patterns in both data types with appropriate uncertainty, suggesting that this is not an issue in this study. Had the model preferentially fit to MGUS prevalence or MM incidence at the expense of the other data type, we agree that a greater exploration of the weighting of the data types would be warranted.

Fitting to MGUS prevalence and validating against MM incidence is an interesting suggestion. Ultimately, we chose to fit to both data types because, fitting only to MGUS prevalence would likely yield more uncertain parameter estimates that would limit the conclusions that we could reach about the contributions of age, sex, and race to MM disparities. Nevertheless, as recommended by Reviewer #1 as well, we have performed an independent validation of our model which can be found in Response 1.5 and in the Supplement on Lines 665-693.

Comment 3.4

Finally, I understand the explanation for the pattern of MGUS duration by age in Figure 4, but I don't understand why the authors could not produce a version of this figure in the absence of other-cause death, which would be a much cleaner comparison of natural history across ages. Can this be added alongside or as an additional figure?

Response 3.4

We thank the reviewer for raising this point. The reviewer is correct that we could generate an expected duration of MGUS by age by taking one over the rate of MM progression. However, this quantity does not account for the competing risk of death. That is, it represents the expected duration of MGUS were an individual not to be subject to all-cause mortality. In practice, the pre-clinical dwell time shown in Fig. 4 is shaped not only by the rate of MM progression but also by all-cause mortality. We show that differences in all-cause mortality drive differences in the pre-clinical dwell time, because the rate of MM progression is not statistically different by race. We believe that this is a contribution of our study that has not been addressed by past studies.

We clarify that the pre-clinical dwell time depends on all-cause mortality in the Methods on Lines 273-274:

“The preclinical dwell time depends upon the rate of progression from MGUS to MM and the competing baseline mortality rate, both of which depend upon age, sex, and race.”

Comment 3.5

The Discussion does not address the limitation of the model structure in that it is essentially age-group-specific and does not lend itself to aging from one age group into another (MM incidence in a given age group depends not only on MGUS prevalence in that group but on MGUS prevalence in prior groups). It would be of value to understand this and to see some discussion of alternative modeling approaches that have been used for cancer natural history modeling e.g. continuous-time multi-state models.

Response 3.5

Our model simulates a discrete-time multi-state model to calculate MGUS prevalence and MM incidence. In this way, it does lend itself to aging from one age group into another. For instance, the prevalence of MGUS in age group a determines the MM incidence in age group $a+1$. We clarify this in the methods on Lines 92-94:

To model the natural history of multiple myeloma¹⁷, we constructed a discrete time, multi-state compartmental model consisting of four health states: healthy (H), monoclonal gammopathy of undetermined significance (MGUS), multiple myeloma (MM), and death (D) (Fig. 1).

We agree with the reviewer that further discussion of the limitations of our approach is warranted. Specifically, we discuss on Lines 432-433 the limitation that we are in effect simulating a single cohort. Future work could build upon ours to account for period effects and time trends across cohorts:

“Additionally, extensions to our modeling framework could simulate multiple cohorts to account for period effects that likely shape MGUS prevalence and MM incidence.”

REVIEWER COMMENTS

Reviewer #1 (Remarks to the Author):

Many thanks to the authors for replying to my concerns.

Ascertainment bias is a significant concern which NHANES and SEER are subject to. This is demonstrated by comparing the MGUS incidence curves in iSTOPMM (<https://ash.confex.com/ash/2022/webprogram/Paper163169.html>) with the corresponding sex/ethnicity matched curves from the authors. This demonstrates a marked underestimation of MGUS in NHANES and SEER and a time-dependent relationship between certain types of MGUS - these will both presumably affect the model?

Reviewer #2 (Remarks to the Author):

The revised manuscript addressed my previous concerns.

Reviewer #3 (Remarks to the Author):

The authors have responded to most of my comments. I would like some further clarification regarding the response to comment 3.4. When the authors say that "differences in all-cause mortality drive differences in the pre-clinical dwell time," this does not make sense under the standard definition of preclinical dwell time. This needs to be clarified. What is the duration of MGUS as graphed in Figure 4? Is it the duration of MGUS conditional on MGUS ending before other-cause death? Please clarify exactly what is being graphed in this Figure. I would still think that a cleaner figure that graphs the duration of MGUS in the absence of other-cause death would be informative because this would just reflect the disease and not the combination of disease natural history and other-cause death.

REVIEWER 1

Comment 1.1

Many thanks to the authors for replying to my concerns. Ascertainment bias is a significant concern which NHANES and SEER are subject to. This is demonstrated by comparing the MGUS incidence curves in iSTOPMM (<https://ash.confex.com/ash/2022/webprogram/Paper163169.html>) with the corresponding sex/ethnicity matched curves from the authors. This demonstrates a marked underestimation of MGUS in NHANES and SEER and a time-dependent relationship between certain types of MGUS - these will both presumably affect the model?

Response 1.1

We thank the reviewer for raising this point. We agree with the reviewer that ascertainment bias may be a concern for these frequently used databases. We used these two databases to estimate MGUS prevalence and MM incidence due to the following reasons.

SEER collects cancer incidence data from population-based cancer registries. These registrars are tested via web-based reliability studies, and audits of high-volume facilities are performed, to ensure that case ascertainment is complete and timely (Yu *et al.*, 2009; PMID: 19418830).

NHANES provides nationally representative data with the unique availability of data from relevant tests for MGUS to allow for the determination of the age-adjusted prevalence by ethnic/racial group (https://wwwn.cdc.gov/Nchs/Nhanes/1999-2000/SSOL_A.htm). MGUS diagnoses in NHANES were confirmed serologically using the FLC assay with sensitivity of 98% (95% CI: 91 – 100%) and specificity of 95% (92 – 98%). Given the high sensitivity and specificity (Katzman *et al.*, 2002; 10.1093/clinchem/48.9.1437) and the fact that blood samples were obtained from a random subset of the NHANES cohort (Landgren *et al.*, 2014), we believe that the potential for underreporting in this dataset is low.

Although the national Icelandic study of MM – iSTOPMM study provides national estimates of MGUS and MM, these estimates are not directly comparable to the U.S. data, because the Icelandic population has a very different demographic makeup – predominantly ethnic Icelanders. As black race is a risk factor for MGUS and MM, we do not believe that the trends in MGUS prevalence between the iSTOPMM and NHANES surveys are directly comparable.

Should the prevalence of MGUS and the incidence of MM be underreported in these databases, we would underestimate the rate of MGUS development and the rate of progression from MGUS to MM. This would similarly lead to an underestimation of the pre-clinical dwell time. We have commented on this in the Discussion on Lines 438-440:

“MGUS prevalence and MM incidence could also be underreported, which would be bias our estimates of the rates of MGUS development and progression from MGUS to MM downward and cause us to underestimate the preclinical dwell time.”

We thank the reviewer for raising the points about bias as we believe that it has strengthened our limitation sections of the Discussion.

REVIEWER 2

Comment 2.1

The revised manuscript addressed my previous concerns.

Response 2.1

We thank the reviewer for reading our revised manuscript.

REVIEWER 3

Comment 3.1

The authors have responded to most of my comments. I would like some further clarification regarding the response to comment 3.4. When the authors say that "differences in all-cause mortality drive differences in the pre-clinical dwell time," this does not make sense under the standard definition of preclinical dwell time. This needs to be clarified. What is the duration of MGUS as graphed in Figure 4? Is it the duration of MGUS conditional on MGUS ending before other-cause death? Please clarify exactly what is being graphed in this Figure. I would still think that a cleaner figure that graphs the duration of MGUS in the absence of other-cause death would be informative because this would just reflect the disease and not the combination of disease natural history and other-cause death.

Response 3.1

We thank the reviewer for taking the time to read our responses. Thanks to the point raised by the reviewer, we now realize that our discussion surrounding the pre-clinical dwell may be misleading and thus have provided further clarification in the revised manuscript. Following Kuntz *et al.* (10.1177/0272989X11408730) and others, we defined the pre-clinical dwell time as the duration of time from the onset of a pre-cancerous state to the onset of a symptomatic cancerous state. Because this definition of the pre-clinical dwell time assumes that the individual survives sufficiently long to develop the symptomatic cancerous state (i.e., MM), it inherently conditions upon the pre-cancerous state (i.e., MGUS) ending before other-cause death. In other words, one can conceptualize two competing risks for individuals with MGUS: (1) progressing to MM or (2) dying from other causes; only those that do not succumb to the competing risk of other-cause death make up the distribution of the pre-clinical dwell time.

To address the reviewer's comment, we have generated a duration of MGUS in the absence of other-cause death by taking the inverse of the rate of progression to MM, as follows:

Had the time scale of MM progression been sufficiently shorter than the time scale of other-cause death (i.e., on the order of days to weeks), then simply taking the inverse of the rate of MM progression and ignoring other-cause mortality would be appropriate. However, the rates of progression to MM and other-cause mortality both occur on similar time scales (i.e., years), so they jointly determine the length of the pre-clinical dwell time. Therefore, it is not possible to isolate the duration of MGUS in the absence of other-cause death.

To clarify this, we have added text to the methods when discussing the pre-clinical dwell time on Lines 276-278:

“This relationship between the preclinical dwell time and the baseline mortality rate exists because individuals with MGUS must survive sufficiently long to progress to MM.”

Additionally, we modified our Discussion of the pre-clinical dwell time on Lines 416-419:

“Because the preclinical dwell time requires that individuals with MGUS survive long enough to develop MGUS, we attribute this phenomenon to two competing effects: (1) the rate of progression from MGUS to MM and (2) the baseline mortality rate.”